# Repurposing Language Models into Embedding Models: Finding the Compute-Optimal Recipe

**Albert Q. Jiang** [*]
University of Cambridge

**Alicja Ziarko** [*]
IDEAS NCBR
University of Warsaw
IMPAN

**Bartosz Piotrowski**
IDEAS NCBR

**Wenda Li**
University of Edinburgh

**Mateja Jamnik** [†]
University of Cambridge

**Piotr Miłoś** [†]
IDEAS NCBR
University of Warsaw
IMPAN, deepsense.ai

## Abstract

Text embeddings are essential for many tasks, such as document retrieval, clustering, and semantic similarity assessment. In this paper, we study how to contrastively train text embedding models in a compute-optimal fashion, given a suite of pre-trained decoder-only language models. Our innovation is an algorithm that produces optimal configurations of model sizes, data quantities, and fine-tuning methods for text-embedding models at different computational budget levels. The resulting recipe, which we obtain through extensive experiments, can be used by practitioners to make informed design choices for their embedding models. Specifically, our findings suggest that full fine-tuning and low-rank adaptation fine-tuning produce optimal models at lower and higher computational budgets respectively.

## 1 Introduction

*Text embeddings* are vector representations of text sequences [Devlin et al., 2019]. A desired property of text embeddings is that their distribution in the embedding space reflects the semantic relations of the embedded texts. This property benefits multiple practical tasks [Muennighoff et al., 2023b], such as document retrieval (where documents are ranked based on the distance between their representations and that of a query), document clustering, sentiment analysis, or textual similarity measurement.

In the era of large language models (LLMs) that are pre-trained on vast textual data, it is a natural idea to capitalise on the language representations learned by these models to achieve high-quality embedding models. A common and effective approach has been to initialise an embedding model from a pre-trained LLM and fine-tune it with a *contrastive loss* [Neelakantan et al., 2022, Izacard et al., 2022]. This phase is

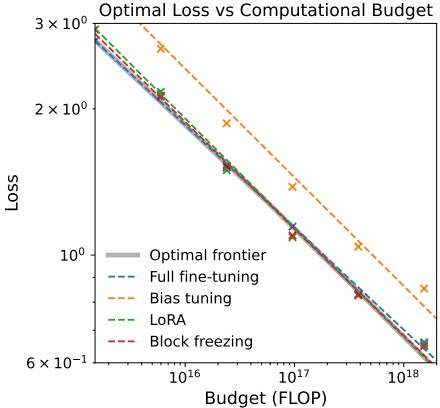

Figure 1: The optimal loss achieved using four different fine-tuning methods (full fine-tuning, only tuning the bias, low-rank adaptation, and freezing transformer blocks) at given budgets. The horizontal axis is the computational budget in floating point operations (FLOP) and the vertical axis is the contrastive loss. The X marks are datapoints and dotted lines are fitted linear trends for different methods. The solid black line is the "optimal frontier," i.e., the optimal loss achievable with a fixed budget and the best method.

---

[*]Equal contribution.
[†]Equal advising contribution.

38th Conference on Neural Information Processing Systems (NeurIPS 2024).

of crucial importance, as the quality of the embeddings extracted directly from a pre-trained LLM (e.g., by averaging hidden states of the last layer) is not sufficient for most tasks.

However, the best performing modern LLMs are usually massive in terms of the parameter counts (e.g., 175B parameters for GPT3 [Brown et al., 2020] and 340B for PaLM2 [Anil et al., 2023]), and are therefore difficult to train with limited resources. This motivates a practical question, that to the best of our knowledge has not been addressed systematically, and thus we address it in this paper: *what is the best embedding model one can train from a backbone decoder-only LLM with a fixed training compute budget*?

To answer this, we performed an extensive empirical study. We started by identifying design choices one can make when fine-tuning language models into embedding models, including: model size, data volume, parameter-efficient fine-tuning technique, and the hyperparameters of the chosen technique. Then, we performed a grid search over the pre-defined design choices, and found the best performing configuration under each computational budget. Using these findings, we established scaling laws for contrastive fine-tuning, which enabled us to build an algorithm that produces a general recipe for efficient fine-tuning of a decoder-only LLM that obtains a high-quality embedding model.

**Contribution** We comprehensively experimented with contrastively fine-tuning language models into text embedding models. We analysed how the choice of model sizes, data quantities, and fine-tuning methods affect the performance in the resource-constrained training regime. We compiled the findings into an algorithm that, given a fixed computational budget, predicts the optimal network architecture, data quantity, and parameter-efficient fine-tuning hyperparameters. We open-source the code to train and evaluate our models at: `https://github.com/SeqDM/Efficient-Embeddings`.

## 2   Related work

**Embedding models** Training neural networks to represent text as continuous vectors was popularised by word2vec [Mikolov et al., 2013], which produced semantically meaningful embeddings of *words*. BERT [Devlin et al., 2019] and the further contrastively trained SimCSE [Gao et al., 2021c] quickly established encoder-only transformers as the go-to architecture for embedding *text*.

More recently, decoder-only transformers have become increasingly more powerful and efficient at the same time [Brown et al., 2020, Touvron et al., 2023, Jiang et al., 2023]. Building an embedding model based on them utilises the knowledge from their pretraining, and thus it is a natural step. Neelakantan et al. [2022] set a successful precedent in this paradigm by initialising embedder training with decoder-only GPT models.

Notably, the current open state-of-the-art (the top open model on the MTEB [Muennighoff et al., 2023b] leaderboard, as of 18 May 2024), `SFR-Embedding-Mistral` [Meng et al., 2024], is also fine-tuned from the decoder-only Mistral 7B [Jiang et al., 2023].

**Benchmarks for embedding models** Embedding models are versatile in their applications. Therefore, a broad and diverse benchmark to provide a robust measure of their performance is called for. The first such benchmark was BEIR introduced by Thakur et al. [2021]. It has nine different information retrieval tasks (e.g., duplicate-question retrieval or citation-prediction) on 18 datasets. Recently, Muennighoff et al. [2023b] introduced MTEB (Massive Text Embedding Benchmark), which is substantially larger than BEIR and, in addition to information retrieval, has seven other types of tasks (e.g., clustering, summarisation, and pair classification) using 58 datasets and covering 112 languages. The authors also provide a leaderboard that currently contains more than 220 entries.

**Scaling laws** Scaling laws predict the performance of models at larger scale (e.g., more parameters, more data) from experiments at smaller scales. Kaplan et al. [2020] and Hoffmann et al. [2022] used empirically fitted laws to predict the performance of neural language models at different model sizes and data scales, making it possible to calculate the optimal configuration before training. Muennighoff et al. [2023a] inspected model training in data-constrained, multi-epoch regime. Frantar et al. [2023] looked at scaling laws for language models in the context of weight sparsity. Zhang et al. [2024] investigates the scaling laws for model fine-tuning, including parameter efficient methods: LoRA [Hu et al., 2022] and prompt tuning [Lester et al., 2021]. They show that the scaling laws and the best fine-tuning method are task dependent. Biderman et al. [2024] investigates the properties of fine-tuning models with LoRA in detail, showing that LoRA underperforms full fine-tuning when fine-tuning the models on mathematical or code datasets.

The single most related investigation is a concurrent work Fang et al. [2024], where the scaling laws for encoder models for retrieval are investigated. There are significant differences in the settings we consider. Our focus is on investigating the process of fine-tuning decoder-only models for good quality embeddings. Moreover, our main goal is to find which strategy for achieving good embeddings is optimal in a budget-restricted settings, while taking into account the popularity of applying parameter efficient methods for fine-tuning, like LoRA or partial model freezing. In spirit, our work is similar to AutoML [He et al., 2021b], which aims to automatically find the optimal ML architecture. However, our method goes beyond and targets the data and the training method as well.

**Parameter-efficient fine-tuning**  Modern language models have a lot of parameters, and fine-tuning them using consumer-grade hardware is difficult in general. Notwithstanding the recent turn to *in-context learning*, fine-tuning is still necessary for most applications. A range of techniques have been developed recently, to make fine-tuning more efficient [Houlsby et al., 2019, Li and Liang, 2021, Qi et al., 2022, He et al., 2021a]. These approaches are all applicable to embedding models. Muennighoff [2022] explored a simple parameter-efficient approach to repurpose GPT models into embedding models where only the bias tensors of the transformer model are updated [Zaken et al., 2022]. [Sun et al., 2024] developed a parameter-efficient method designed specifically for fine-tuning embedding models. However, a systematic study of what parameter-efficient methods are optimal under what scenarios has not been performed, which is the aim of our work.

## 3 Preliminaries

### 3.1 Scaling laws and compute-optimal models

The constraint optimisation problem that we aim to solve is the following: *given a fine-tuning corpus and a family of pre-trained decoder-only language models of different sizes, minimise the contrastive loss subject to a fixed computational budget.*

Following Hoffmann et al. [2022, Section 3.2], we perform a hyperparameter search at different computational budget levels to obtain IsoFLOP profiles of models. We then find the optimal models at each level, fit a loss-size scaling law, and extrapolate it to unobserved FLOP budgets. We only focus on contrastive fine-tuning since state-of-the-art text embedder models are all trained in such fashion [Gao et al., 2021c, Neelakantan et al., 2022, Wang et al., 2023].

We refer to the models with the lowest contrastive loss achievable with fixed computational costs as *compute-optimal models*. We focus on optimising three components specifically: model size, data quantity, and fine-tuning method. For each fine-tuning method, we find the optimal model configuration at each FLOP level and establish the relationship between the computational budget and optimal loss for that method. Then, for any unobserved computational budget, we use the scaling laws to predict the loss, and pick the method that gives the lowest loss and its corresponding configuration.

### 3.2 Extracting representations from transformers

We start with a standard decoder-only transformer model architecture, such as GPT architectures [Radford et al., 2019, Brown et al., 2020], or Pythia [Biderman et al., 2023]. Given a sequence of tokens $S = (s_1, \ldots, s_n)$, we pass them through the transformer model and average the last layer representations as the embedding vector $\text{emb}(S) = \frac{1}{n} \sum_{i=1}^{n} h_i \in \mathbb{R}^m$, where $h_i$ is the last layer hidden state of the token $s_i$.

Some works apply the last token pooling approach [Neelakantan et al., 2022] instead of using mean pooling described above. Here, however, we default to mean pooling as in [Wang et al., 2022, Li et al., 2023, Günther et al., 2023]. It is a more popular method overall, and moreover, we find it to yield better performance, which we demonstrate in Appendix E.3.

### 3.3 Contrastive fine-tuning

We use the contrastive loss as in [Neelakantan et al., 2022], which has been widely used in fine-tuning pre-trained language models [Wang et al., 2022, 2023, Günther et al., 2023].

For a batch of $n$ examples, that is, $n$ pairs of texts $(x_1, y_1), \ldots, (x_n, y_n)$, we only treat examples $(x_i, y_i)_{i=1}^{n}$ as positive matches, and all the pairs $(x_i, y_j)$, where $i \neq j$ as negatives. Concretely, we calculate the batch contrastive loss in two steps. First, we calculate the logits:

$$\texttt{logits[i,j]} = \texttt{sim}(\texttt{emb}(x_i), \texttt{emb}(y_j)) \cdot \exp(\tau),$$

where `sim` is the cosine similarity function between two vectors, and $\tau \in \mathbb{R}$ is the temperature. The total loss is the sum of the cross entropy losses on both the row and the column directions:[1]

```
labels = [0, 1, ..., n - 1]
l_r = cross_entropy(logits, labels)
l_c = cross_entropy(logits.T, labels)
loss = (l_r + l_c) / 2
```

**Hard negatives in contrastive learning**  In our contrastive learning setting, we use in-batch examples as negatives. Some datasets additionally include tailored negative examples (*hard negatives*) for each positive datapoint, which can be incorporated into the contrastive loss to promote learning more precise representations. Moreover, there are works that focus on approaches for mining hard negatives which result in better training data in the context of specific downstream tasks [Xiong et al., 2021, Zhan et al., 2021].

However, recent works aiming at training powerful, general-purpose embedding models that do rely on datasets with hard negatives often arrive at embeddings by having two distinct training phases: the expensive "unsupervised" *phase one* involving vast data from the internet, and then a "supervised" *phase two*, often targeted towards a specific downstream task, where training data of lower quantity – but containing high-quality hard negatives – is used [Li et al., 2023, Wang et al., 2022, Xiao et al., 2023]. Since the first phase is more expensive, it will benefit more from scaling laws such as the ones we derive.

Moreover, the usefulness of hard negatives highly depends on their quality, which may vary wildly between datasets. Therefore, conclusions reached with hard negatives are more closely tied to the datasets the models are trained on. Since we develop scaling laws agnostic to the dataset, we abstain from using hard negatives in our experiments.

## 3.4 Fine-tuning methods

The most basic and straightforward method of fine-tuning is *full fine-tuning*, where *all* the weights of the model are updated under the contrastive objective.

We also study parameter-efficient fine-tuning (PEFT) methods, which are popular, especially in academic settings, because they have lower memory requirements than full fine-tuning. Since the PEFT methods update fewer parameters in the network than there are in total, the relationship between their computational cost, model size, and data quantity is different from full fine-tuning. Hence, we individually study the IsoFLOP profiles for four fine-tuning methods, namely, *full fine-tuning*, *block freezing*, *bias-only tuning*, and *Low-Rank Adaptation (LoRA)*. Each of the last three non-standard methods is briefly characterised below.

**Block freezing**  In this approach, we *freeze* the LLM token-embedding layer as well as the first $k$ transformer blocks so that their parameters stay fixed during fine-tuning. Only the latter part of the model is back-propagated through and updates its parameters. To process the same amount of data, this is more computationally efficient than full fine-tuning. By varying the number of frozen blocks, one trades-off between the computational efficiency and flexibility of the model.

**Low-rank adaptation (LoRA)**  This method was introduced by Hu et al. [2022] to significantly reduce the number of trainable parameters of a neural network while maintaining high performance. LoRA introduces a small number of weights into the model and only trains those. It can be applied to a subset of dense layers of a model. It works by parametrising each dense layer $W$ from a subset of linear layers of a model by two low-rank matrices $A$ and $B$, and using $W + AB$ instead of $W$, while training only $A$ and $B$.

**Bias-only tuning**  In this method, only bias parameters are fine-tuned, and the rest of the model remains frozen. This approach was proposed by Zaken et al. [2022] as a parameter-efficient fine-tuning method for BERT encoders, and recently applied by Muennighoff [2022] to contrastively fine-tune GPT models.

## 3.5 Calculating computational cost

We denote the number of non-token-embedding parameters used in the forward pass to process each token as $N_F$, the number of non-token-embedding parameters in backward-propagation as $N_B$, the num-

---

[1]We adopt the PyTorch notation here, where the cross entropy loss takes two arguments: the logits of the probability distribution, and the indices of the true categories.

ber of non-token-embedding parameters updated during backward-propagation as $N_U$, the total number of tokens processed as $D$, and the computational cost as $C$ floating point operations (FLOP). Following the calculation by Kaplan et al. [2020], we derive the relationship between the variables to be

$$C = 2N_F D + 2N_B D + 2N_U D.$$

In case of full fine-tuning, every parameter is trainable, so $N_F = N_B = N_U$, and therefore

$$C = 6N_F D.$$

This is the same as the formula for the computational cost used by Kaplan et al. [2020] for pre-training.

## 4 Experiments

We first specify the relevant details of our experimental setup (Section 4.1). Next, we present the results of our experiments where we contrastively train a grid of models of different sizes, using different computational budgets, and apply different compute-optimal fine-tuning methods with varying hyperparameters (Section 4.2). Based on the collected data, we fit a mathematical formula describing the observed scaling laws (Section 4.3). Finally, we provide general observations and recommendations for efficient contrastive fine-tuning (Section 4.6). In these experiments[2] we investigate the compute-constrained setup, but we present limited results on the data-constrained setup in Appendix D.

### 4.1 Experimental setup

We use eight decoder-only models from the Pythia suite [Biderman et al., 2023], which have sizes 14M, 31M, 70M, 160M, 410M, 1B, 1.4B, and 2.8B of parameters. All the models have been pre-trained on the Pile dataset [Gao et al., 2021a] for 300 billion tokens. We consider six computational budgets equally spaced on the log-scale: 1.5e15, 6e15, 2.4e16, 9.6e16, 3.8e17, and 1.5e18 FLOP.

We fine-tune our models on the English partition of the BAAI BGE dataset [Xiao et al., 2023], which contains 200 million semantically related (*query*, *value*) pairs from various internet sources such as Wikipedia and Stack Exchange. Note that since the BAAI BGE dataset is massive, for all our experiments we fine-tune for less than one epoch on this corpus, which allows us to avoid the effect of diminishing returns on subsequent training epochs [Muennighoff et al., 2023a].

We use the AdamW optimiser [Loshchilov and Hutter, 2019] and a cosine learning rate scheduler during training. The learning rate first goes through a linear warm-up phase of $1/10$ of the total steps to a peak that is $1/10$ of the pre-training peak learning rate, that is, to learning rates between $1.2 \times 10^{-5}$ and $6 \times 10^{-5}$. Then, it decays in a cosine schedule to $1/10$ of the maximum at the end of training. We employ gradient checkpointing [Chen et al., 2016] and GradCache [Gao et al., 2021b] to limit the peak GPU RAM usage during training. We use the Transformers [Wolf et al., 2020] library for the training of the models and use the Accelerate [Gugger et al., 2022] library to facilitate multi-GPU training. The batch size in our main line of experiments is 1024 sequences and the context length is 75 tokens. We give the full list of training hyperparameters in Appendix E.

We also perform a less extensive grid of experiments to confirm that our findings are robust to change in hyperparameters. We find that our scaling law formula fitted to losses from models with batch size 1024 and context length of 75 tokens describes the loss for models trained with batch size 2048 and context length of 512 tokens very well. The detailed description and plots are shown in Appendix G.

In addition to controlling the final training contrastive loss achieved by the models, we also measure downstream performance by evaluating the models on a representative subset of the MTEB benchmark [Muennighoff et al., 2023b]. The benchmark defines eight categories of tasks in total (e.g., retrieval, semantic text similarity). We select one task from each category by determining which ones are the most correlated with the performance for the whole category (see Appendix B for details).

### 4.2 Experimental results for different methods

**Full fine-tuning** With full fine-tuning, the numbers of forward and backward parameters are the same, and the computational cost is straightforwardly $C = 6N_F D$. In Figure 2a, we present the final contrastive loss vs. the number of parameters in the network on a log-log scale.

---

[2]The data collected in our experiments and their interactive visualisations are available in this Colab notebook: `https://colab.research.google.com/drive/1EC2QdVrOIXhamLJI51CUywWJ33sApXX8`

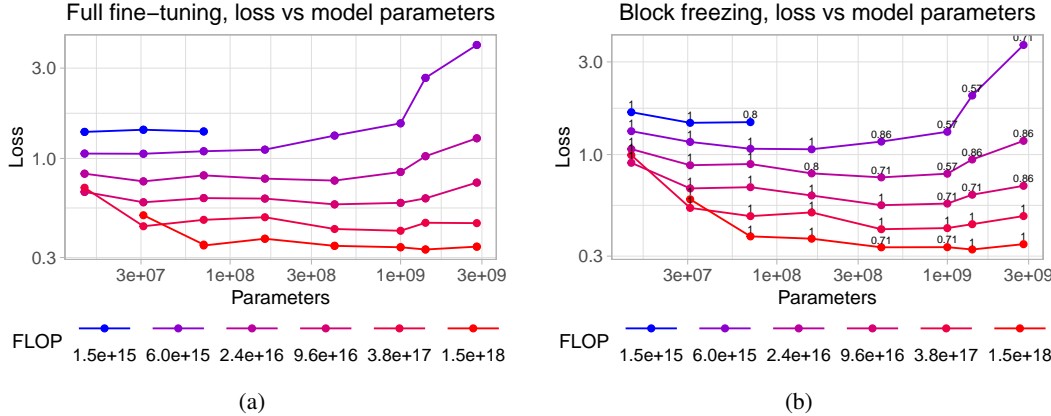

(a)

(b)

Figure 2: **(a)** IsoFLOP profiles for *full fine-tuning*. The horizontal axis is the number of parameters in the model, and the vertical axis is the achieved loss. Both axes use log-scale. The optimal model size tends to increase as the computational budget increases. **(b)** IsoFLOP profiles for *block freezing*. The axes are the same as for full fine-tuning. Each data point denotes the optimal choice with respect to the fraction of active blocks during training, which is noted above the points. The optimal model size tends to increase as the computational budget increases, while the optimal active block fraction tends to slightly decrease as the model size gets larger.

As expected, larger computational budgets consistently improve the loss. The IsoFLOP profiles resemble the classical ones from Hoffmann et al. [2022]. However, they tend to be flatter when the model size and computational budget are both small or both large. From the IsoFLOP profiles we can see what is the optimal model size that minimises the loss at each computational budget.

**Block freezing** Since the Pythia models we experimented with have $6, 12, 16, 24$ or $32$ blocks, we freeze $\frac{0}{6}, \frac{1}{6}, \ldots, \frac{5}{6}$ of their blocks if their number is divisible by 6, and $\frac{0}{8}, \frac{1}{8}, \ldots, \frac{7}{8}$ if it is divisible by 8. In this setup, unlike for full fine-tuning, the token-embedding parameters always remain frozen even when all the transformer blocks are active. We have $N_B = N_U < N_F$, and hence cost $C = 2N_F D + 4N_B D$.

In Figure 2b, we present IsoFLOP profiles for optimal loss at given model sizes. The trend of the profiles is quite similar to that of full fine-tuning presented in Figure 2a.

We plot the loss with varying model sizes, computational budgets, and fractions of frozen blocks in Figure 3. For smaller models ($\leq 160M$ parameters), a lower fraction of frozen blocks leads to a lower final loss across all computational budgets. However, for larger models, using more than $70\%$ of transformer often only marginally improves the model. For instance, for the model with $1e9$ parameters, it is visible that freezing $50 - 70\%$ of the blocks gives the optimal results for all budgets except for $3.8e17$ FLOP.

**Bias-only tuning** Here we simply update only the bias parameters and not the weights of the model. Figure 4a shows the final training loss across all the model sizes and computational budgets. Increasing the computational budget decreases the optimal loss that is achievable, although the absolute values of the losses are high compared to other fine-tuning methods (the lowest achievable loss is above 0.4 with bias-only tuning).

**Low-rank adaptation** We use the LoRA implementation from the PEFT library [Mangrulkar et al., 2022]. We attach the adapter to all the dense layers of the network, and vary the adapter rank from $8$ up to $2048$, since even lower ranks resulted in an unstable training.

We compare the loss obtained for each model size and computational budget for different LoRA rank values in Figure 5. In almost all the combinations of the model size and computational budget, the rank of $32$ or $128$ (occasionally $512$) is optimal. Wang et al. [2023] also use LoRA in the context of contrastive learning, but they fix the rank of $16$ for their experiments. Our findings suggest that this choice should be reconsidered as being potentially suboptimal.

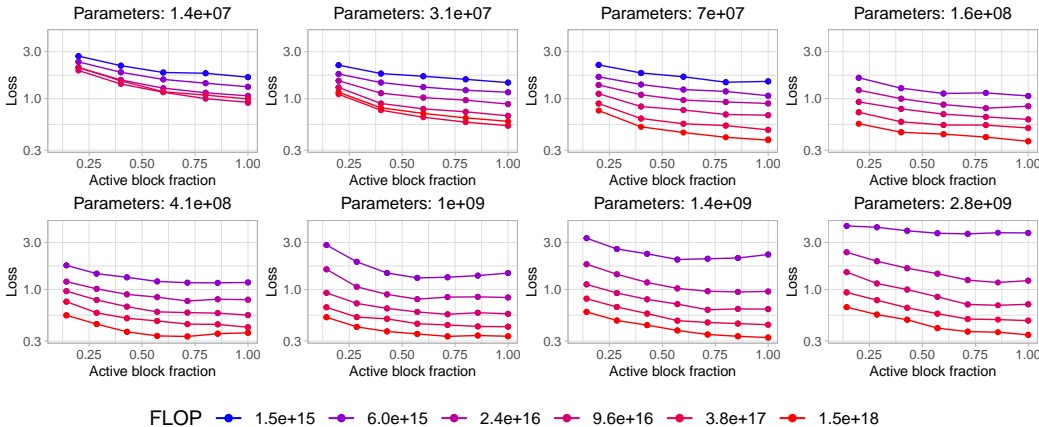

Figure 3: The effect of block freezing across all model sizes. Different colours signify different computational budgets. Unless the model is large and the computational budget small, it is always better to update all the (non-embedding) weights of the model.

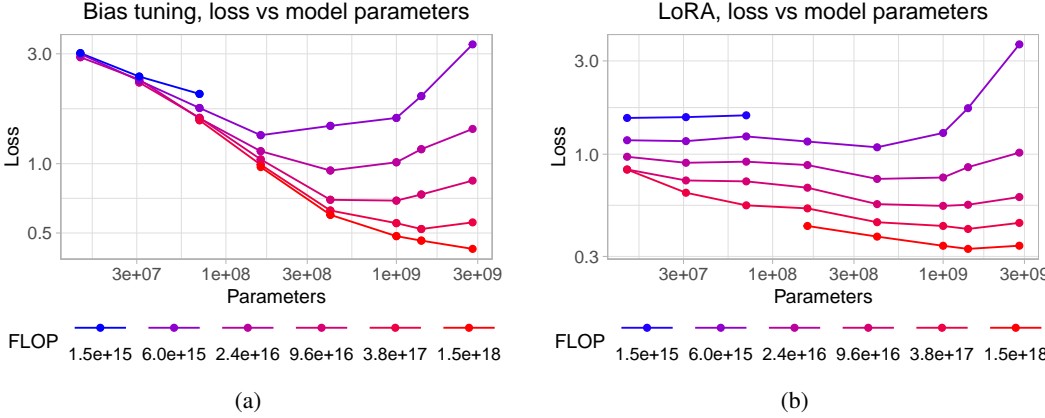

Figure 4: **(a)** IsoFLOP profiles for *bias-only tuning*. The horizontal axis is the number of parameters in the model, and the vertical axis is the achieved loss. Both axes use log-scale. The optimal model size increases as the computational budget increases, but the achievable loss is higher than for other fine-tuning methods. **(b)** IsoFLOP profiles for *LoRA* fine-tuning. The axes are the same as for bias-only tuning. Each data point denotes the optimal choice of the rank of LoRA matrices given the size of the model and the computational budget.

In Figure 4b, we present IsoFLOP profiles for LoRA, where each data point represents the best LoRA rank setting for a given model size and computational budget. We see that the relationship between the optimal loss and the LoRA rank is quite consistent across model sizes and computational budgets, meaning that the rank hyperparameter is not very sensitive with respect to the two variables. In practice, this means that setting the LoRA rank to 32 or 128 are likely to be good default choices.

**Lookup table of optimal model sizes for different methods** In Figure 6a (for full fine-tuning), Figure 6b (for LoRA), and Figure 13 (for freezing and bias tuning, in Appendix A), we visualise the relationship between the optimal model size and the computational budget. This makes looking up the optimal configuration for a given method at a given budget very convenient.

**Downstream performance** In Figures 8, 9, 10, and 11 (in Appendix A) we present the IsoFLOP plots of the downstream performance for all the four fine-tuning methods on the subset of the MTEB benchmark (described in Appendix B). Based on the performance results, we also present the optimal size choices per budget in Figure 15 in Appendix A (which is a performance analogue of Figure 13), and the optimal performance plot in Figure 12 (a performance analogue of Figure 1). These are mostly consistent with the loss results, albeit more noisy. In Appendix C, we demonstrate the significant correlation between the final training loss and the downstream task performance.

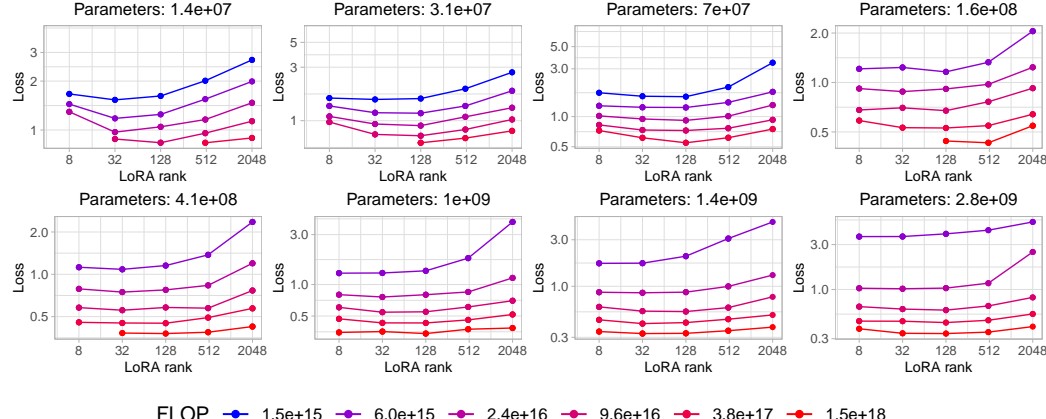

Figure 5: The effect of different LoRA ranks across all model sizes. Different colours signify different computational budgets. The inflected curves indicate that it is less beneficial to use a rank from either extremes of the spectrum (8 or 2048). The detrimental effect of the high rank of 2048 is stronger for lower computational budgets. Ranks of 32 and 128 result in the lowest loss overall.

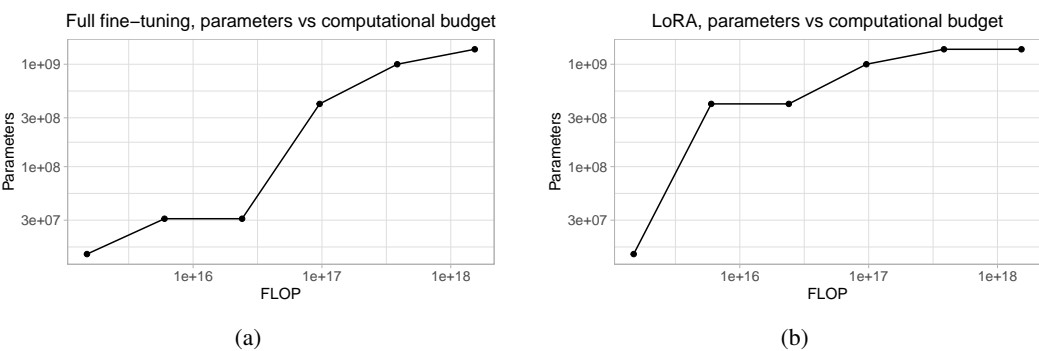

Figure 6: Optimal model size vs. computational budget for **(a)** full fine-tuning, and **(b)** LoRA.

## 4.3 Scaling laws for embeddings

We aim to model the behaviour of the loss $L$ with an analytical formula as a function of the number of parameters $N$ and the number of training tokens $D$. Following Hoffmann et al. [2022], we start with the following formula:

$$L(N, D) = C + \frac{A}{N^\alpha} + \frac{B}{D^\beta},$$

where $A, B, C, \alpha, \beta \in \mathbb{R}_+$ are real-valued parameters to be fitted. We found that this formula reasonably describes the loss behaviour for each of our methods. However, the fraction of *trainable parameters* is an important factor that is not considered in the model above. Following the modelling of Frantar et al. [2023], we propose a modified formula:

$$L(S, N, D) = C + \frac{a_d \log(D) + b_d}{N^\alpha} + \frac{a_s(1 - S)^{b_s} + c_s}{D^\beta},$$

where $S$ is an additional variable representing the trainable fraction of parameters, and $a_d, b_d, a_s, b_s, c_s \in \mathbb{R}$ are additional coefficients to be fitted. We check that this new formula fits the data better than the original one, and conjecture that it can be used for larger models and larger computational budgets. More details about the derivation are presented in Appendix F.

## 4.4 Compute-optimal frontier and recipe

In Figure 1, we plotted the best achievable losses against the computational budgets for the four fine-tuning methods considered. We then fitted a linear trend for each method on a log-log scale (equivalent to a power law relationship on a normal scale), and highlighted the lowest loss achievable at given

---

**Algorithm 1** Recipe for compute-optimal embedding model

---

**Input:** compute budget $C$.
**Output:** fine-tuning method, model size, data quantity, and (optionally) the method's hyperparams.
**if** $B \leq 9.06\text{e}16$ FLOP **then**
    Use full fine-tuning.
    Go to Figure 6a to find the optimal model parameters $N$ given budget $C$.
    Calculate the data quantity $D$.
    **return** Full fine-tuning, $N$, $D$, ().
**else**
    Use LoRA.
    Go to Figure 6b to find the optimal model parameters $N$ given budget $C$.
    Go to Figure 5 to find the LoRA rank $R$ according to $C$ and $N$.
    **return** LoRA, $N$, $D$, $(R)$.

---

budgets using the best corresponding fine-tuning method. We call the contour of this function that predicts the optimal loss across methods the compute-optimal frontier.

The equation describing the full fine-tuning and the LoRA linear fits are

$$\ln(\text{loss}) = -0.21 \cdot \ln(\text{budget}) + 8.39, \text{ and } \ln(\text{loss}) = -0.22 \cdot \ln(\text{budget}) + 8.93,$$

respectively. The equations suggest that at a budget lower than $9.06\text{e}16$ FLOP, the optimal model is achieved with full fine-tuning, and at a higher budget, the optimal model is achieved with LoRA.

With this, one becomes fully equipped to deduce the optimal recipe for text embedding model training for a given budget, by carrying out the procedure detailed in Algorithm 1. Note that although the block freezing approach is not predicted to be optimal at any budget, its performance is quite close to optimal. It also has the advantage of being easy to implement and has lower memory requirements, hence might be a good method to choose in practical use cases for larger budgets. Its optimal model size and fraction of active blocks hyperparameter can be worked out in the same way as for the other methods, with the help of Figure 3 and Figure 13 (left pane) in Appendix A.

We briefly discuss the case of data-optimal frontier in Appendix D. We do not emphasise this setup due to it being less standard in language modelling research.

### 4.5 Generalisation

To assess the generality of our findings, we investigate whether our analytical formula for predicting loss—fitted to datapoints from the Pythia suite (from Section 4.3) can also describe the behaviour of models from a different family. We conduct experiments on two language models, namely Gemma 2B [Mesnard et al., 2024] and Gemma 2 2B Team et al. [2024]. We trained both models with the computational budgets of $1.5\text{e}15$, $6\text{e}15$, $2.4\text{e}16$, $9.6\text{e}16$, $3.8\text{e}17$, and $1.5\text{e}18$ FLOP.

Figure 7 presents results for Gemma 2B with both full fine-tuning and LoRA fine-tuning, while results for Gemma 2 2B are shown in Appendix A. In both cases, the formula provides a close approximation of the loss behavior. Although this study is limited to a single model size, it offers an initial indication that our findings may generalize to new models.

### 4.6 Takeaways

We compile the takeaways from our extensive experiments by analysing each fine-tuning method individually and contrasting them afterwards.

- Bias-only tuning is not a good fine-tuning strategy for embedding model training, as it is consistently worse than other strategies under IsoFLOP comparisons.
- LoRA and block freezing are both effective approaches that improve the performance of the trained embedding models at higher budgets. For LoRA, the rank hyperparameter is not very sensitive to model size or computational budgets, with an optimum at around 128.
- Finding the optimal recipe for fine-tuning embedding models requires a non-trivial effort, and the resulting optimal configurations are sometimes counter-intuitive (e.g., not using the largest model size possible). Carefully tuning hyperparameters is crucial and should be practised systematically.

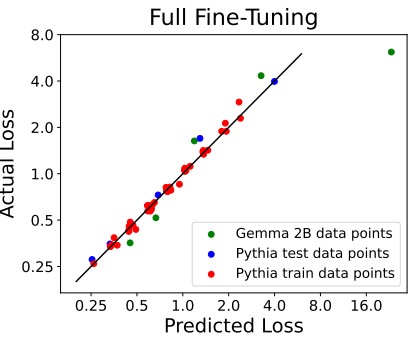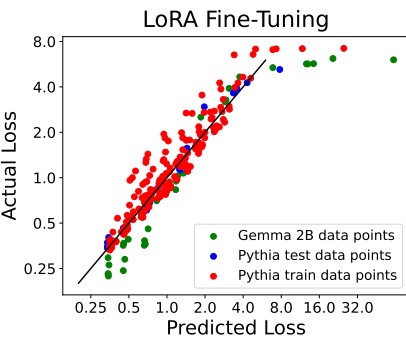

Figure 7: Actual loss vs. predicted loss for full fine-tuning and LoRA finetuning. Red points are the models from runs on Pythia suite of size smaller than 2.8B, blue points are from runs on Pythia 2.8B and green points are from runs on Gemma 2B. The formula is fitted using the red points.

## 5   Limitations and future work

**Other families of models**   In this paper, we experimented with the Pythia [Biderman et al., 2023] suite of pre-trained models, which are of different sizes but have all been trained on 300B tokens. This means that they have been over-trained with respect to the Chinchilla scaling law [Hoffmann et al., 2022] to different degrees. The relationship between the optimal model size and the computational budget might differ for a suite of pre-trained models with different over-training ratios. We note that prior efforts [Dettmers et al., 2024] have found that fine-tuning conclusions reached with Pythia do generalise to other model families such as OPT [Zhang et al., 2022], LLaMA [Touvron et al., 2023], and BLOOM [Scao et al., 2022], in the quantised setting. Indeed, in Section 4.5, we find that a model from a different family (Gemma [Mesnard et al., 2024]) closely fits our loss predictions.

**Averaging**   Due to constraints on our computational resources, we ran each experiment exactly once. Averaging over multiple random seeds for the experiments would reduce the variance involved in each, and potentially result in less noisy conclusions. This is a straightforward extension that we leave to carry out in future work, should we have more resources. We also plan to evaluate the trained models on the entirety of the MTEB benchmark [Muennighoff et al., 2023b], instead of using the losses from the last steps of training, to better reflect the performances of embedding models.

**Other forms of embedding readout**   We only experimented with mean pooling as a way of extracting embeddings from the transformer models. Other readout methods could be used, like the max pooling or extracting the hidden state corresponding to the end-of-sequence token, which might result in different scaling laws and optimal frontiers.

**Inference cost analysis**   In this work, we only focus on the training cost and refrain from taking inference costs into account. However, in certain practical scenarios, the latter may also be relevant.

## 6   Conclusions

In this paper, we systematically investigated the influence of pre-trained model size, fine-tuning data quantity, and fine-tuning method in the final performance of embedding models repurposed from language models. As a result, we devised an algorithm that takes in the computational budget and returns the optimal configuration for the embedding model to train. This enables researchers who wish to adapt language models to embed their own data, especially those with limited computational resources, to obtain optimal text embedding models with greater efficiency in time and resources.

## Acknowledgements

The authors would like to thank Denis Kuznedelev, Soroush Tabesh and Konrad Staniszewski for helpful discussions and feedback. AQJ is supported by a Peterhouse graduate studentship. PM was supported by National Science Center Poland under the grant agreement 2019/35/O/ST6/03464. We gratefully acknowledge Polish high-performance computing infrastructure PLGrid (HPC Centers: ACK Cyfronet AGH) for providing computer facilities and support within computational grant no. PLG/2023/016717.

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

# A   Additional results

In Figure 8 (for full fine-tuning), Figure 9 (for block freezing), Figure 10 (for LoRA), and Figure 11 (for bias tuning), we present the IsoFLOP profiles where downstream task performance is shown. The downstream performance is measured on a subset of MTEB tasks described in Appendix B.

Using the location of optimal loss given the computational budget for freezing and bias tuning methods – as illustrated in Figures 2b and 4a – we project optimal model sizes for each computational budget in Figure 13. Similar figures for full fine-tuning and LoRA are presented in the main text in Figure 6a and Figure 6b, respectively.

Using the location of *optimal downstream task performance* given the computational budget for all the four fine-tuning methods – as illustrated in Figures 8, 9, 10, and 11 – we project optimal model sizes for each computational budget in Figure 15.

In Figure 14 we present how our formula fitted to models from the Pythia suite describes the loss behaviour of the Gemma 2 model.

# B   Downstream task evaluation details

For each category present in the MTEB benchmark [Muennighoff et al., 2023b], we take the data from the models evaluated in the original paper and select the task that has the highest correlation with the category average. This results in 8 tasks from the MTEB benchmark used for the downstream task evaluation:

1. EmotionClassification (classification),
2. TwentyNewsgroupsClustering (clustering),
3. SprintDuplicateQuestions (pair classification),
4. AskUbuntuDupQuestions (re-ranking),
5. SciFact (retrieval),
6. STS15 (semantic text similarity)
7. SummEval (summarization),
8. Tatoeba (bitext mining).

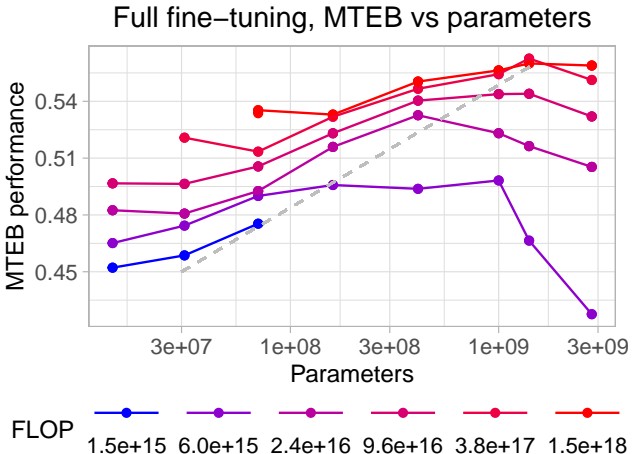

Figure 8: IsoFLOP profiles for *full fine-tuning*, where performance on MTEB tasks is observed instead of the loss. Both axes use log-scale. The performance remains approximately inversely correlated with the loss. However, interestingly, there are no corresponding plateaus on the IsoFLOP profiles.

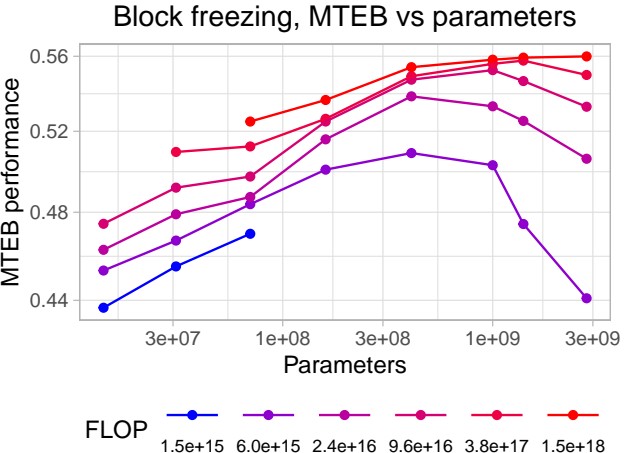

Figure 9: IsoFLOP profiles for *freezing*, where performance on MTEB tasks is observed instead of the loss. For a given FLOP level, the model with the optimal active block fraction is depicted.

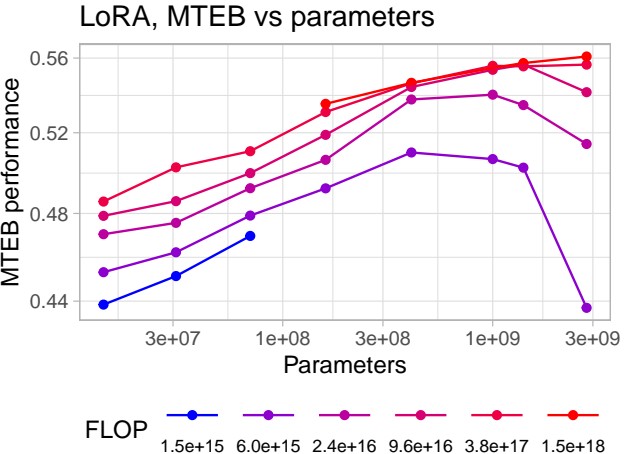

Figure 10: IsoFLOP profiles for *LoRA*, where performance on MTEB tasks is observed instead of the loss. For a given FLOP level, the model with the LoRA rank is depicted.

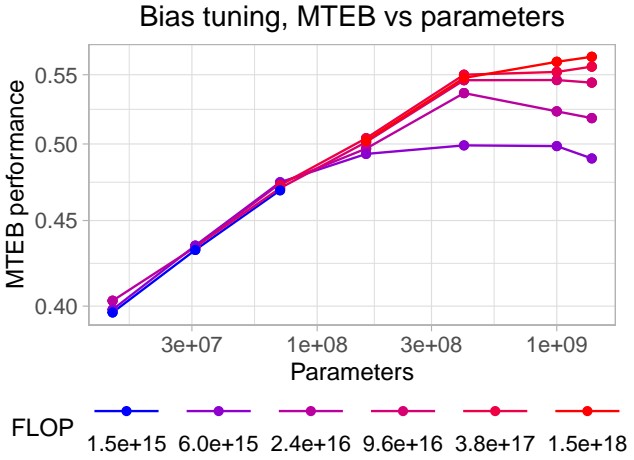

Figure 11: IsoFLOP profiles for *bias tuning*, where performance on MTEB tasks is observed instead of the loss.

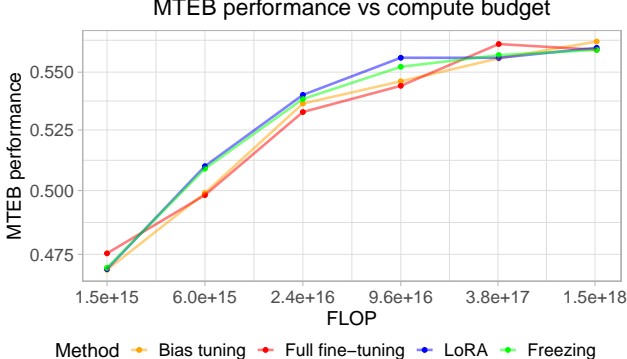

Figure 12: The optimal downstream task performance achieved using all four different fine-tuning methods at given budgets. This is a performance analogue of Figure 1 (which shows final training loss instead).

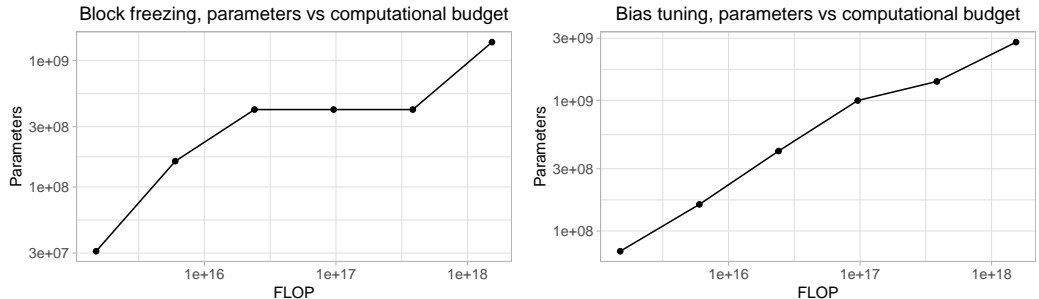

Figure 13: Optimal choices of model size given the computational budget, with respect to the *final training loss*, for block freezing and bias tuning.

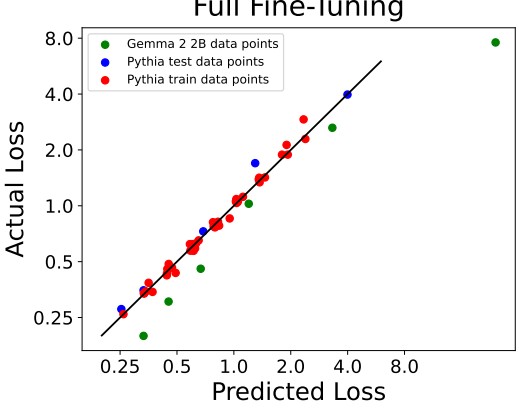

Figure 14: Actual loss vs. predicted loss for full fine-tuning and LoRA finetuning. Red points are the models from runs on Pythia suite of size smaller than 2.8B, blue points are from runs on Pythia 2.8B and green points are from runs on Gemma 2 2B. The formula is fitted using the red points.

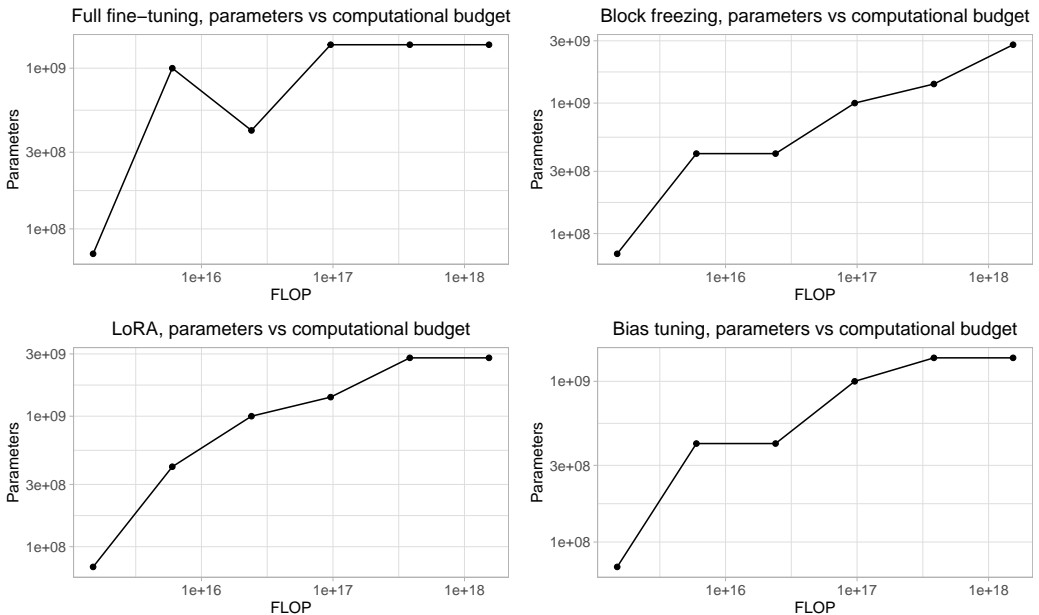

Figure 15: Optimal choices of model size given the computational budget, with respect to the *performance on the downstream task evaluation.*

For each of the task above, we specify its category in brackets. Since Tatoeba is not featured in the MTEB leaderboard,[3] we skip it when calculating our approximation of the MTEB performance of the embedding models.

In principle, one could evaluate all the checkpoints on all the MTEB tasks. However, we decided to evaluate on a representative subset in order to speed up evaluation (evaluating some MTEB task takes more than 24 hours, and we have close to 1000 checkpoints, so the overall cost is significant).

## C  Correlation between loss and MTEB performance

In our work, we use loss as the main metric used to formulate the conclusions in form of the compute-optimal recipe and the scaling laws. This is in line with other works investigating scaling laws in various contexts, like [Hoffmann et al., 2022, Muennighoff et al., 2023a, Zhang et al., 2024]. However, we additionally evaluated the trained models on a subset of MTEB benchmark. We calculated the Spearman's rank correlation between the loss and the MTEB performance and it is equal to $-0.892$, which is significant. Additionally, Figure 16 visually compares the loss and MTEB performance.

## D  Data-Constrained Setup

In our work, we mainly focus on finding the compute-optimal recipe. The training datapoints we share can however be utilised to obtain conclusions about other setups. In particular, we can consider the data-constrained setup. Here, we compare models trained with the same amount of training data instead of the same amount of computation. In Figure 17 we present the plot the final contrastive loss vs. the amount of tokens seen by the network on a log-log scale. The conclusion that can be drawn is that in a data-constrained setup, one should use the largest model available and fine-tune it using full fine-tuning or LoRA with a rank on the bigger side.

## E  Hyperparameters for the training setup

Below, we list the hyperparameters that we used for training.

---

[3]`https://huggingface.co/spaces/mteb/leaderboard`

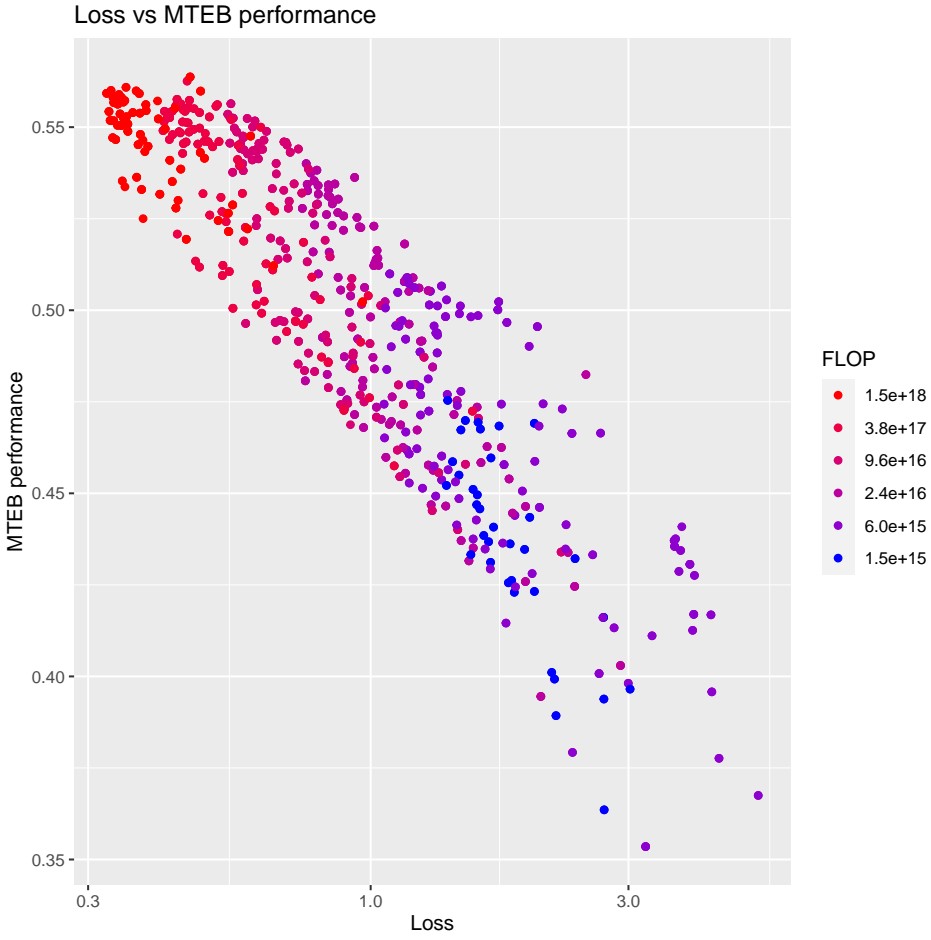

Figure 16: All the final checkpoints involved in our suite of experiments, evaluated bath in terms of the final training loss and the performance on a representative subset of MTEB benchmark. The correlation between these two metrics is significant.

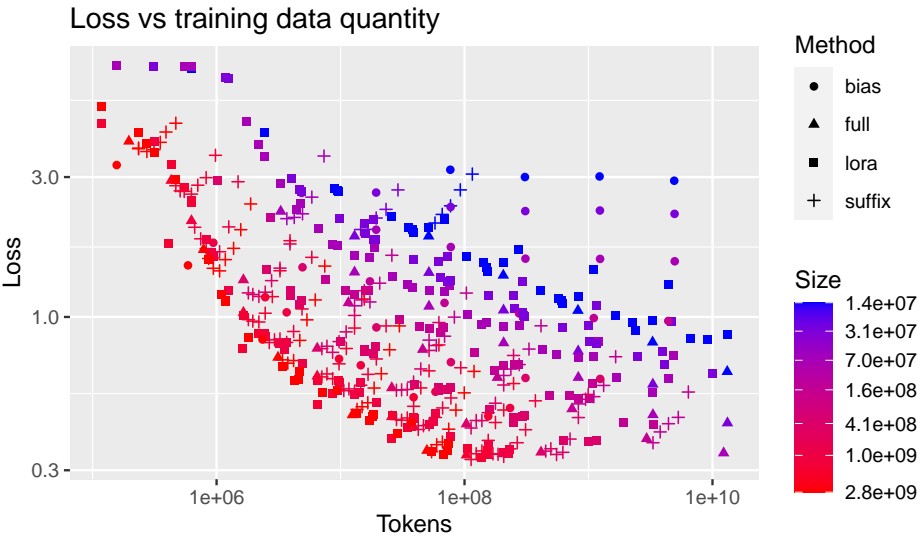

Figure 17: Final training loss vs number of training tokens for all the models. It is visible that given the number of training tokens, the best performance is obtained by the largest model.

```
batch_size = 1024
context_length = 75
AdamW.weight_decay = 0.1
tau = 0.025 # temperature parameter
```

## E.1  Learning rate for block freezing and full fine-tuning

For all the experiments with block freezing and full fine-tuning, we base our peak learning rate on the peak learning rate from Pythia [Biderman et al., 2023]. We calculate our learning rate by dividing the Pythia learning rate by ten. We use the cosine learning rate schedule with warmup, warmup period being fixed to 10% of the training steps. Learning rate starts at zero, goes up to the peak learning rate, then follows the cosine schedule back to zero.

## E.2  Temperature and weight decay for partial tuning

We found the temperature to affect the training performance rather significantly, while the weight decay was of a very slight importance. We run a grid search for partial tuning, varying temperature and weight decay and arrived at a choice of $\tau = \frac{1}{40}$ and weight decay $0.1$. We compared the runs by calculating the results on a subset of MTEB benchmark. Grid results for temperature are detailed in Table 1, with best scores for each of the temperature value listed.

Table 1: A comparison of different temperature values

| Temperature$^{-1}$ | Avg. score |
|---|---|
| 50 | 0.5 |
| **40** | **0.51** |
| 35 | 0.5 |
| 30 | 0.5 |
| 10 | 0.47 |
| 1 | 0.3 |

## E.3  Readout: last vs average

We found that using the averaging of embeddings as the pooling method results in a better performance over selecting the last embedding. We compare the results of the best run for mean and last pooling on full MTEB without MSMarco subset. The scores are presented in Table 2.

Table 2: Mean pooling works better than last pooling.

| Pooling method | Avg. score |
|---|---|
| **Mean pooling** | 0.47 |
| Last pooling | 0.36 |

## E.4  LoRA hyperparameters

In order to be able to compare the loss from LoRA with partial block freezing, we leave most of the hyperparameters unchanged. We change the learning rate, we find the optimal learning rate by running a grid search over hyperparameters. The result shows that for 3 different rank choices, the same learning rate works best. The exact values are presented in Table 3.

## E.5  Bias tuning hyperparameters

Similarly, as with LoRA, for bias tuning to be comparable with partial block freezing, we leave most of the hyperparameters the same. We vary the learning rate, we find the optimal learning rate by running a grid search over hyperparameters. The exact values can be seen in Table 4.

Table 3: Learning rate for LoRA.

| Learning rate | LoRA rank | Loss |
|---|---|---|
| 1e2 | | 1.96 |
| 1e3 | 8 | 1.35 |
| 1e4 | | 2.22 |
| 1e2 | | 1.96 |
| 1e3 | 16 | 1.34 |
| 1e4 | | 2.22 |
| 1e2 | | 2.09 |
| 1e3 | 32 | 1.32 |
| 1e4 | | 2.19 |

Table 4: Learning rate for bias tuning.

| Learning rate | Loss |
|---|---|
| 1e2 | 1.24 |
| 1e3 | 1.37 |
| 1e4 | 2.97 |

# F  Scaling laws

We aim to express the formula describing the behaviour of the loss for our models as a function of the number of parameters $N$ and training tokens $D$. Following Hoffmann et al. [2022] we start with:

$$L(N, D) = C + \frac{A}{N^\alpha} + \frac{B}{D^\beta},$$

where $A, B, C, \alpha, \beta \in \mathbb{R}$ are parameters to be fitted based on the data.

Our setting differs from that of Hoffmann et al. [2022] as we train using the contrastive objective instead of the next-token prediction task, and moreover we apply various compute-efficient methods making some fraction of the parameters of the model *not trainable*. Despite that, we can still observe certain linear trends that the loss follows. For all the methods, when the FLOP budget is not overly restricted, if we fix the amount of both *all* the parameters as well as the *trainable* parameters and vary the amount of FLOP, the loss behaves approximately linearly. We observe that for full fine-tuning, the points are roughly collinear, as in Figure 18. On the other hand, for LoRA and partial block freezing, we discover that while the lines for one model are parallel, they are translated, depending on the amount of trainable parameters, which can be observed in Figures 21 and 22.

Following the Frantar et al. [2023]'s approach, we replace the term $\frac{B}{D^\beta}$ by $\frac{(a_s(1-S)^{b_s}+c_s}{D^\beta}$, where $S$ is an additional parameter indicating the fraction of the model's parameters that are trainable, and $a_s, b_s$ are real-valued constants. Similarly, if we keep the forward token amount fixed and vary the amount of model parameters, the linear trend can be observed. Again, the lines are parallel and differ by an intercept, that seems to be linearly dependent on the logarithm of forward tokens, shown in Figure 20. Therefore, we replace the term $\frac{A}{N^\beta}$ by $\frac{a_d \cdot \log(D) + b_d}{N^\beta}$, where $a_d, b_d$ are real-valued constants. The final formula describing the loss in our setting has the form:

$$L(S, N, D) = C + \frac{a_d \cdot \log(D) + b_d}{N^\alpha} + \frac{a_s \cdot (1 - S)^{b_s} + c_s}{D^\beta}.$$

We split our dataset into train and test set, with test set being the points corresponding to Pythia 2.8B which is the biggest model that we consider and train set the rest. Similarly to Hoffmann et al. [2022], we use L-BGFS optimization algorithm and Huber loss. We find that $\delta = 0.001$ prevents over-fitting and decreasing it further does not bring more benefits. We then compare the base formula being fitted formula from Chinchilla to our modified formula and find that our works better, as demonstrated in Figure 19. Both formulas are fitted after a grid search on the initial values of their coefficients. It can

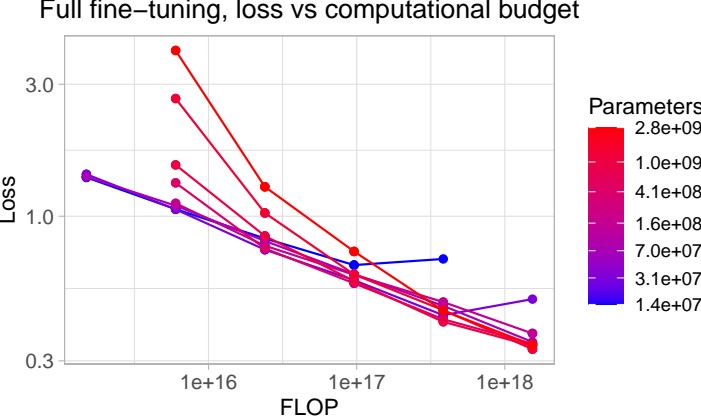

Figure 18: Loss on the $y$ axis and FLOP on the $x$ axis. Lines of the same colours correspond to the same amount of parameters in the model. Excluding points corresponding to severely constrained FLOP budget, the loss behaviour can be approximated by straight lines.

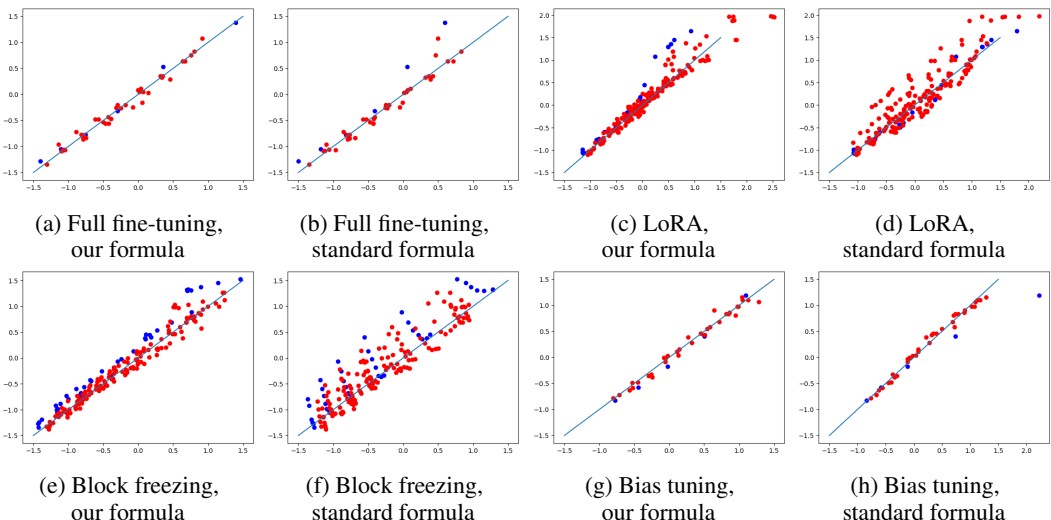

| | |
|---|---|
| (a) Full fine-tuning, our formula | (b) Full fine-tuning, standard formula |
| (c) LoRA, our formula | (d) LoRA, standard formula |
| (e) Block freezing, our formula | (f) Block freezing, standard formula |
| (g) Bias tuning, our formula | (h) Bias tuning, standard formula |

Figure 19: The actual loss on the $y$ axis and predicted loss on the $x$ axis. Red points are the training set and blue the test set. Figures 19a, 19c, 19e, and 19g are the results of fitting our formula for different fine-tuning methods, while Figures 19b, 19d, 19f, and 19h are the result of fitting the standard formula from Hoffmann et al. [2022]. It can be observed, that in contrast to the standard formula, our results in plots that are roughly linear, without additional trends visible.

be observed that we are able to accurately predict the loss for the bigger model for all the methods. Therefore, we hypothetize that our formula can be used for prediction the performance of models of an even larger scale.

## G   Robustness to increase in batch size and context length

Constrained by the computational budget, we performed the biggest line of experiments using batch size 1024 and context length 75. However, an increased context length as well as batch size might be desirable in many cases. Here, we demonstrate that the scaling laws established by us extend to a setting with batch size 2048 and context length 512. For those experiments, we trained five pre-trained decoder-only models from the Pythia suite [Biderman et al., 2023], with sizes 14M, 31M, 70M, 160M, and 410M. We consider five computational budgets: 1.5e15, 6e15, 2.4e16, and 9.6e16 FLOP. We fit the formula following the method from Appendix F, with the change of using the

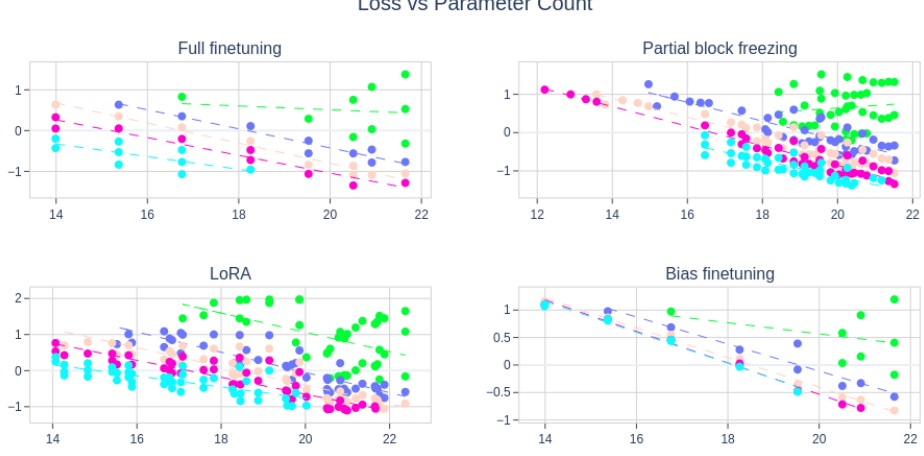

Figure 20: Logarithm of loss on the $y$ axis and logarithm of the parameter count on the $x$ axis. The colors represent different quantiles of the amount of tokens used in the model fine-tuning. Linear dependency can be observed, with the trends being approximately parallel for different quantiles.

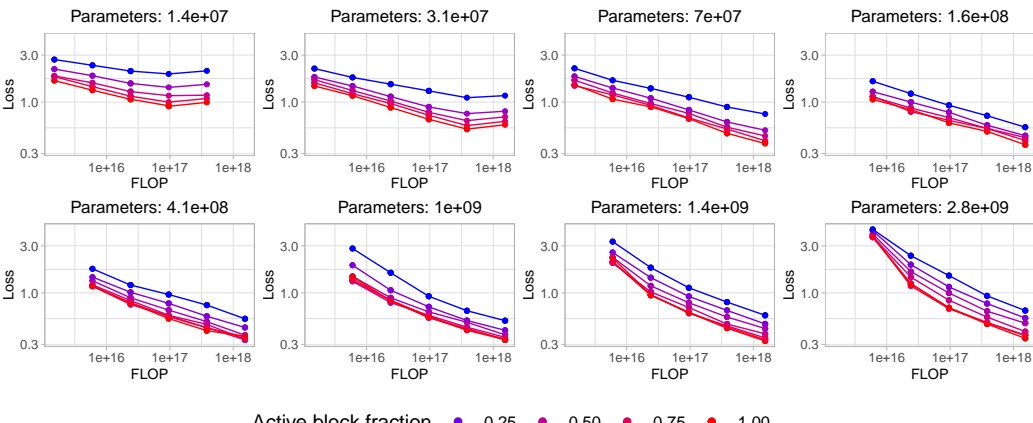

Figure 21: Plots of losses for block freezing. Each subplot contains datapoints corresponding to one model size. Loss on the $y$ axis and FLOP on the $x$ axis. Lines of the same colours correspond to the same amount of trainable parameters in the model. The loss behaviour can be approximated by straight lines.

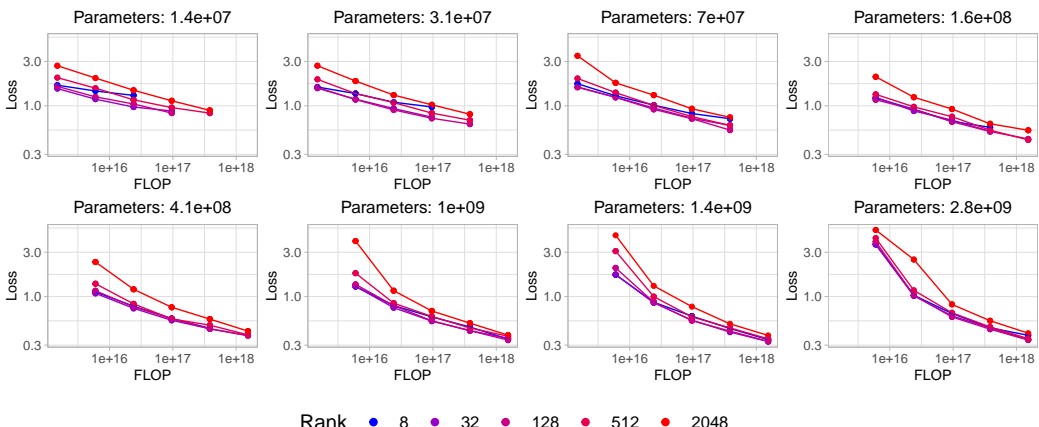

Rank ● 8 ● 32 ● 128 ● 512 ● 2048

Figure 22: Plots of losses for LoRA. Each subplot contains datapoints corresponding to one model size. Loss on the $y$ axis and FLOP on the $x$ axis. Lines of the same colours correspond to the same amount of trainable parameters in the model. The loss behaviour can be approximated by straight lines.

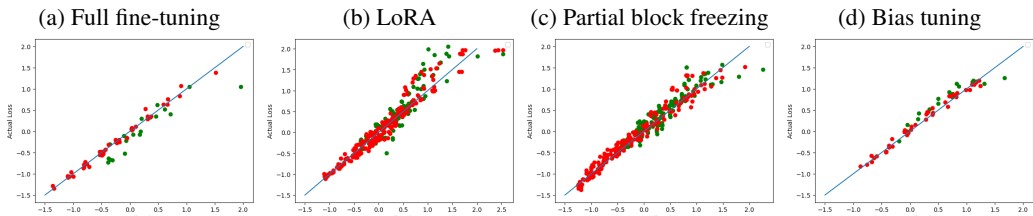

Figure 23: Plots of predicted vs actual losses for the 4 finetuning methods. The formula is fitted on the data coming from trainings with batch size 1024 and context length 75 (red points). It is then used to predict the losses for data points trained with batch size 2048 and context length 512 (green points).

number of training steps as the measure of data seen by the model instead of forward tokens. We then use it to predict the loss for the changed hyperparameters values. The results are visible in Figure 23. We conclude that our findings are relevant for setups with a larger batch size and context length.

# H   Computational resources

For all the trainings, we used a cluster with A100 GPUs with 40GB VRAM or with 80GB VRAM. We used 4 CPUs and 128GB of RAM per GPU. We estimate that the results contained in this article were obtained by expanding somewhere in between 30k and 40k GPU hours.

# I   Dataset

We utilize the BAAI dataset available at `https://data.baai.ac.cn/details/BAAI-MTP`.

It has a custom licence that allows for conducting academic research.

