# OpenReview forum: "Repurposing Language Models into Embedding Models: Finding the Compute-Optimal Recipe"
_NeurIPS.cc/2024/Conference — NeurIPS 2024 poster_

### Official Review · Reviewer_xguk · 2024-07-12

**Soundness:** 3
**Presentation:** 2
**Contribution:** 3
**Rating:** 7
**Confidence:** 3

**Summary:**

The paper fine-tunes Pythia LLMs with different sizes for information retrieval (IR) applications. The paper investigates the effect of different model sizes, FLOPs, fine-tuning methods such as LoRA or bias tuning, and some important hyperparameters (e.g., dimension in LoRA) on loss and downstream applications (measured by MTEB). Some takeaways are interesting and counterintuitive (e.g., the larger model does not necessarily achieve better loss).

**Strengths:**

This paper conducts very extensive experiments and analyses in the main paper and appendix. I appreciate all the efforts and believe that it would be very useful for practitioners who want to create their own embedding models from LLM.

**Weaknesses:**

Although I think the experiments in this paper are valuable, I think main goal of the analyses is not very practical. Personally, I think the term Compute-Optimal from Chinchilla (https://arxiv.org/pdf/2203.15556) and this paper is very misleading because it only considers the training cost and ignores the inference cost. In practice, the LLMs are often trained much longer than this "optimal" value. For example, Chinchilla-optimal for 8B LLM is just 200B tokens, but LLaMA 3 is trained using over 15T tokens (https://ai.meta.com/blog/meta-llama-3/). For the information retrieval (IR) models, the inference cost might be even more important compared to LLMs.

In addition to main complaint above, I think that some details could be explained more clearly (see the questions below). Finally, I think that the figures with FLOPs on the x-axis and different curves are for different model size are much easier to read than what the paper has now (the model size on the x-axis and different curves are for different FLOPs), but it might be my personally preference.

**Questions:**

1. Although you show that the correlation between MTEB and loss is very high (−0.892) overall, some important trends seem to be different, which might affect your takeaway message. For example, the best score of MTEB always comes from the largest model in Figure 7-10, but the lowest loss often comes from smaller model size in Figure 2 and 4. The bias tuning methods are significantly worse in terms of loss, but it seems to achieve the best MTEB when FLOP is 5e+18 in Figure 11. When you write your takeaways, why not also include the results of MTEB?

2. I don't understand l_r and l_c in line 134 and 135. Why don't we need to use labels.T for l_c?  If this is a standard loss, please cite. Otherwise, please provide more explanations/motivations for this loss.

3. Why do you choose to use the English partition of the BAAI BGE dataset, which is designed for Chinese embedding? If it is not a common choice, you should explain more about this.

4. Did you get the performance of full MTEB? Does the full MTEB give you different trends? In the figures, MTEB performances means the performances on the task subset, right? I think you should make it clear in those figures (MTEB performances -> MTEB subset performances).

**Limitations:**

The current limitation section is good and adding some discussion of inference cost could make it better.

---

> ### Author Rebuttal · Authors · 2024-08-07
>
> We thank the Reviewer for their insightful feedback! Addressing your questions and concerns:
>
> **(W1) The term Compute-Optimal [...] is very misleading because it only considers the training cost and ignores the inference cost [which is important in practice].**
>
> We do agree that our usage of the “compute-optimal” term is somewhat imprecise – we have updated the paper, pointing out that we mean the training cost only. We have also added an appropriate paragraph in the limitations section.
>
> We believe that there is value in deriving the functional relationship between the model size, training method, data, and the final training loss (which is also done in many previous works on scaling laws). This relationship clearly characterises the ingredients needed to achieve particular performances. We refrain from taking inference costs into account, as they are hard to estimate. However, we stress that by having the functional form, the reader can analyse their particular use case, which may take into account the sum of the training and inference cost as a constraint, and find the optimal iso-total-cost models.
>
> **(W2) I think that the figures with FLOPs on the x-axis and different curves are for different model size are much easier to read than what the paper has now [...]**
>
> Thank you for this suggestion – to increase the readability, we have added plots where the x-axis is the model size, not FLOPs. During experimentation we were looking at both versions and decided that it is better to present loss vs size plots as the IsoFLOP profiles in them directly answer our motivating question: given the FLOP budget, what is the optimal model size one should use?
>
> **(Q1) Although you show that the correlation between MTEB and loss is very high (−0.892) overall, some important trends seem to be different, which might affect your takeaway message. [...]**
>
> We thank the Reviewer for raising this. Ultimately, the most suitable metric to use is the one most relevant to the reader. We believe people who have the need to fine-tune their own embedding models and the resources to do so should use their own benchmarks to derive the scaling laws most accurate for them. Without access to the relevant benchmark, we find the validation loss to be the smoothest metric to base scaling laws upon (again, this approach is typically assumed in the scaling laws works). The MTEB results have higher variances and do not necessarily align with the user’s requirement.
>
> **(Q2) I don't understand `l_r` and `l_c` in line 134 and 135. Why don't we need to use `labels.T` for `l_c`? If this is a standard loss, please cite. [...]**
>
> It is a standard symmetric contrastive loss used throughout the contrastive learning literature for training textual embedding models [3,4,5,6] (and in other applications [7]). The specific formulation included in our paper originates from [3] (Section 2.2). We cite this work (please see line 126), but we will emphasise it more to avoid confusion – thank you for your suggestion!
>
> As for the question about `labels.T` for `l_c`: it should remain `labels` – we just use PyTorch notation here; the first argument to the cross entropy function is a matrix, and the second argument is a vector indicating for each row what is the index of the “correct” element. (We link the documentation in the footnote on page 3.)
>
> **(Q3) Why do you choose to use the English partition of the BAAI BGE dataset, which is designed for Chinese embedding? [...]**
>
> The reason for choosing BAAI BGE is that this dataset is a massive collection of data for training embeddings. Since we do not consider scaling laws under data constraints like [1], we needed a large enough dataset on which none of our experiments would exceed 1 epoch of training; BAAI BGE was the only published dataset satisfying our constraints.
>
> As for why we used only the English partition of BAAI BGE: we want to explore a family of decoder-only models with different model parameter scales, and the Pythia family is the only available option at the time of experiments that offers lots of sub-billion-parameter models, which are common for embedding models. The Pythia family was pre-trained on The Pile [2], which is an English-only dataset, and hence it does not have Chinese knowledge and capabilities. Therefore, we only focused on the English partition of BAAI BGE.
>
> We have updated the paper to include the full explanation provided here.
>
> **(Q4) Did you get the performance of full MTEB? Does the full MTEB give you different trends? [...] you should make it clear in those figures [that only a subset of MTEB was used].**
>
> The Reviewer is correct here—for practical computational purposes, we did use a subset of MTEB instead of the whole benchmark. (As we explain in Appendix B, this subset was selected in a principled way to be representative of the whole MTEB.)
>
> During the rebuttal period, we checked that the results on the whole suite (except MSMARCO, which is the most expensive to run) are highly correlated with our selected subset. To this end, we compared the two evaluations for 8 checkpoints from the full fine-tuning experiment, and the Pearson correlation is 0.976. This information has been included in the revised version of the paper. We have also fixed the captions to “MTEB subset performance.”
>
> ### References
>
> [1] Muennighoff et al.: Scaling data-constrained language models. NeurIPS 2023\
> [2] Gao et al.: The Pile: An 800GB dataset of diverse text for language modeling. arXiv 2021\
> [3] Neelakantan et al.: Text and Code Embeddings by Contrastive Pre-Training. arXiv 2022\
> [4] Wang et al.: Text Embeddings by Weakly-Supervised Contrastive Pre-training. arXiv 2022\
> [5] Wang et al.: Improving Text Embeddings with Large Language Models. arXiv 2024\
> [6] Günther et al.: Jina Embeddings: A Novel Set of High-Performance Sentence Embedding. arXiv 2023\
> [7] Eysenbach et al.: Inference via Interpolation: Contrastive Representations Provably Enable Planning and Inference. arXiv 2024

---

> ### Comment · Reviewer_xguk · 2024-08-07
> **Thank you for your detailed explaination**
>
> The authors address my concerns well. I have no other major reasons to reject this paper, so I raise my score from 6 to 7.

---

> > ### Author Response · Authors · 2024-08-09
> > **Thank you**
> >
> > We thank the reviewer for their prompt response and for increasing the score of our manuscript.
> >
> > We are also happy to answer new questions if any arise.
> >
> > Best regards, Authors

---

### Official Review · Reviewer_zJJS · 2024-07-13

**Soundness:** 3
**Presentation:** 3
**Contribution:** 3
**Rating:** 7
**Confidence:** 4

**Summary:**

This paper proposes a new algorithm to automatically find optimal configurations of model sizes, data quantities, and fine-tuning methods for text embedding models at different computational budget levels by contrastive pretraining. Specifically, the paper analyzes the choices of different configurations and then designs an algorithm to predict optimal architecture based on budget. The paper also designs a new task to find optimal model configuration given limited budgets. The paper first introduces basics, including the scaling laws, the constrained optimization problem, the contrastive learning,  fine-tuning methods, and then computational cost formulas. The paper then experiments with different finetuning methods to find the optimal loss regarding parameters. Furthermore, the paper aims to find the relationship between loss functions regarding the number of parameters (trainable/total) and the number of training tokens. The paper then fit the plots to find the best equations.

**Strengths:**

1. The paper proposes a new computation optimization problem: given a fine-tuning corpus and a family of pre-trained decoder-only language models of different sizes, minimize the contrastive loss subject to a fixed computational budget. The newly proposed problem is important and has the potential to expand.
2. The paper conducts a comprehensive analysis of existing fine-tuning techniques with different hyperparameters. The paper visualizes the results with clear figures and concludes with several useful results. The paper also predicts the optimal loss for different budgets. The paper evaluates the downstream results.
3. The paper provides code, additional evaluation results and analysis in the Appendix, and detailed computational resources.

**Weaknesses:**

1.  The paper's generalization ability is limited. As mentioned in the limitation, the proposed computation law is only on the Pythia; additional language models might better reflect the generalization ability. The paper also only focuses on contrastive finetuning.
2. The whole paper mainly focuses on the performance analysis. It would be better to include additional theoretical support for the phenomenon that appears in the paper. The paper can also be strengthened by testing the proposed performance law on some unobserved configurations.
3. The paper claims to also analyze the relationships between the data quantities and the performance. However, I cannot find it in Section 4.2. The paper fails to follow the NeurIPS citation format, which uses numbers for references/citations.

**Questions:**

See weakness

**Limitations:**

The paper includes a limitation section in Section 5.

---

> ### Author Rebuttal · Authors · 2024-08-07
>
> We thank the Reviewer for their effort to assess our work! Below we address each of the concerns.
>
> **(1a) The paper's generalization ability is limited. As mentioned in the limitation, the proposed computation law is only on the Pythia; additional language models might better reflect the generalization ability.**
>
> We agree that having additional models would allow for a better reflection of the paper’s generalisation. During the rebuttal we ran experiments with a very recent model – Gemma 2b [1], using LoRA and full fine-tuning methods.
>
> Importantly, we found that the **the loss formula with parameters fitted to the Pythia data points, adequately describes the behaviour of Gemma 2b training for the ‘medium’ budgets**.
>
> At the extremely low computational budget, Gemma performs better than the prediction, which intuitively is caused by the general higher model capability than those in the Pythia suite; also, scaling laws tend to be much less regular in smaller regimes – see Figure 4 in the pdf attachment.
>
> As a result of being time-restricted, we did not perform the two highest budget trainings, but we believe that the results so far are a strong indication as to the generalisability of our findings. We will update our paper to contain those new findings once we have the complete results for Gemma.
>
> **(1b) The paper also only focuses on contrastive finetuning.**
>
> We intentionally focus only on contrastive learning: this has become a standard approach to obtaining useful embedding models. We note that there has been no study before ours that helps to select the most compute-efficient variant of contrastive learning approaches in decoder-only models, rendering our study a worthwhile contribution.
>
> **(2) The whole paper mainly focuses on the performance analysis. It would be better to include additional theoretical support for the phenomenon that appears in the paper. The paper can also be strengthened by testing the proposed performance law on some unobserved configurations.**
>
> We provide theoretical support by utilising a classical risk decomposition model (similar to that of the Compute-Optimal Scaling Laws paper [2]), and adapting it to account for the difference in forward and backward computation parameters in Section 4.3. With this theoretical framework we describe the loss behaviour in terms of the model parameters (both forward and backward), and the amount of data.
>
> We also show that the scaling law generalises to the very recent Gemma 2B model (see above).
>
> **(3a) The paper claims to also analyse the relationships between the data quantities and the performance. However, I cannot find it in Section 4.2.**
>
> In this paper, we focused on investigating the compute-bounded setting – the data-quantity factor is only present implicitly as it can be inferred from the FLOP limit and the model size. However, we now realize that the data-centric perspective is also very valuable and we should analyse it explicitly, especially in the light of the claim in line 157. For this reason, we *attach Figure 3* to our 1-page pdf attachment with plots. A simple and practical conclusion can be inferred from it – if the data is restricted, one should choose the biggest model available and use full fine-tuning or LoRA with a rank on the bigger side. We have updated our paper to contain this plot and conclusion.
>
> We note that these conclusions can be achieved using the raw experimental data that we provide – please, see the link at the bottom of page 5 of our paper.
>
> **(3b) The paper fails to follow the NeurIPS citation format [...].**
>
> Thank you for noticing it! This has already been fixed.
>
> ### References
>
> [1] Mesnard et al.: Gemma: Open Models Based on Gemini Research and Technology. arXiv 2024\
> [2] Hoffmann et al.: Training compute-optimal large language models. arXiv 2022
>
> ---
> *We are grateful for the Reviewer’s feedback and hope the concerns are appropriately addressed. If that is the case, we kindly ask you to reconsider the paper score.*

---

> > ### Comment · Reviewer_zJJS · 2024-08-11
> >
> > Thank you very much for your detailed explanation! The authors have addressed my concern. I will raise my score to 7.

---

> ### Author Response · Authors · 2024-08-11
> **Thank you**
>
> We again thank you for your feedback, and we are grateful for your support of our work!

---

### Official Review · Reviewer_4Bw9 · 2024-07-17

**Soundness:** 2
**Presentation:** 3
**Contribution:** 2
**Rating:** 5
**Confidence:** 3

**Summary:**

This paper investigates how to effectively train text embedding models from pre-trained decoder-only language models while considering computational budget constraints. The authors explore the influence of model sizes, fine-tuning methods, and computational budgets on the performance of the embedding models. The results demonstrate that full fine-tuning is optimal for lower computational budgets, whereas low-rank adaptation fine-tuning is more effective for higher computational budgets.

**Strengths:**

(1) This paper focuses on the important problem of finding the optimal training settings for embedding models within a fixed compute budget.

(2) This paper conducts extensive experiments across 8 model sizes, 6 compute budgets, and 4 tuning methods.

(3) Although the paper uses contrastive loss as the primary measure of model performance, the authors show a strong correlation between contrastive loss and downstream task performance.

**Weaknesses:**

(1) In L51-52, it is stated that "given a fixed computational budget, predicts the optimal network architecture, data quantity, and parameter-efficient fine-tuning hyperparameters". However, data quantity, a very crucial factor, is not investigated in the paper.

(2) The conclusion that full fine-tuning and low-rank adaptation fine-tuning produce optimal models at lower and higher computational budgets respectively seems somewhat superficial.

(3) I have concerns about the selection of the MTEB subset. Retrieval is a crucial task for which embedding models can be used. However, only SciFact is evaluated. Scifact is not a representative retrieval dataset as its corpus is relatively small (only with 5183 documents).

(4) Figure 1 is somewhat hard to read since it is difficult to distinguish the lines representing full fine-tuning, LoRA, and block freezing.

(5) The average pooling method to extract representation is not entirely reasonable, as the autoregressive characteristic would cause the outputs of the first tokens to lack information about the latter tokens.

(6) L141-146 is not entirely accurate. Hard negative mining methods like [1] and [2] do not require this two-stage training procedure. This two-stage training procedure is more of a recent trend in training powerful general-purpose embedding models.

[1] Lee Xiong, Chenyan Xiong, Ye Li, Kwok-Fung Tang, Jialin Liu, Paul Bennett, Junaid Ahmed, Arnold Overwijk. Approximate Nearest Neighbor Negative Contrastive Learning for Dense Text Retrieval. ICLR 2021.

[2] Jingtao Zhan, Jiaxin Mao, Yiqun Liu, Jiafeng Guo, Min Zhang, Shaoping Ma. Optimizing Dense Retrieval Model Training with Hard Negatives. SIGIR 2021.

**Questions:**

Please see my comments in Weakness.

**Limitations:**

The paper has a proper Limitations and future work section.

---

> ### Author Rebuttal · Authors · 2024-08-07
>
> We thank the Reviewer for detailed and informative feedback. Below we address each of the concerns.
>
> **(1) [Despite you mention so in lines 51-52] data quantity, a very crucial factor, is not investigated in the paper.**
>
> In this paper, we focused on investigating the compute-bounded setting – the data-quantity factor is present implicitly as it can be inferred from the FLOP limit and the model size. Following your comment, we recognise that the data-centric perspective is also very valuable. For this reason, we *attach Figure 2* to our 1-page pdf file with plots. A simple and practical conclusion can be inferred from it – if the data is restricted, one should choose the biggest model available and use full fine-tuning or LoRA with a rank on the bigger side. We have updated our paper to contain this plot and conclusion.
>
> We note that these conclusions can be achieved using the raw experimental data that we provide – please, see the link at the bottom of page 5 of our paper.
>
> **(2) The conclusion that full fine-tuning and low-rank adaptation fine-tuning produce optimal models at lower and higher computational budgets respectively seems somewhat superficial.**
>
> We do agree with the Reviewer that our study did not reveal patterns that are unexpected or peculiar. Nevertheless, we argue that our findings constitute a solid empirical grounding and, thus a concrete scientific contribution to helping researchers and practitioners working with embedding models. It is the first study devoted to establishing compute-efficient approaches to contrastive fine-tuning of decoder-only language models. Given the recent activity in this area, we found it to be a serious research gap that needed to be addressed.
>
> Also, please note that our study concerns not only the selection of the optimal method from amongst the four pre-selected, but we also go deeper and analyse important hyper-parameters of those methods (LoRA rank and active block fraction). This was not previously studied, with a notable exception of limited analysis in [2].
>
> **(3) [...] Retrieval is a crucial task for which embedding models can be used. However, only SciFact is evaluated. Scifact is not a representative retrieval dataset as its corpus is relatively small [...].**
>
> Our selection of SciFact was guided by its strong correlation (0.927) with the overall performance in the retrieval tasks category, as evidenced by the Table 11 of MTEB [8]. Nevertheless, we concur with the Reviewer's observation regarding the size of the SciFact benchmark.
>
> Therefore, we now run new experiments to investigate the *correlations* between the performance on SciFact and two large retrieval benchmarks: *Natural Questions* [6] (2.6M test data points), and *DBPedia* [7] (4.6M test data points) for our best models for each (FLOP, method) combination, which are *0.976* and *0.971* respectively (also see Figure 3 of the attachment). We hence conclude, that despite its humble size, SciFact is a satisfactory representative of the retrieval category. This clarification has been added to the paper.
>
> **(4) Figure 1 is somewhat hard to read since it is difficult to distinguish the lines representing full fine-tuning, LoRA, and block freezing.**
>
> We thank the Reviewer for the suggestion. We have updated the figure to make it more legible in Figure 1 of the attachment.
>
> **(5) The average pooling method to extract representation is not entirely reasonable, as the autoregressive characteristic would cause the outputs of the first tokens to lack information about the latter tokens.**
>
> While using the average pooling method may indeed seem unintuitive, it works very well in practice. In *Appendix D, section 3*, we compare the two most popular methods for extracting embeddings – average and last pooling methods – for Pythia 410m, fine-tuned with the FLOP budget of 9.6e+16. We perform evaluation on the whole MTEB [8] without MSMARCO, and we find the difference in performance is 0.47 vs 0.36 in favour of average pooling.
>
> We note that average pooling has been used in several previous works [1,2,3]. Moreover, theoretically, the average pooling method has the possibility to converge to the ‘last’ method by zeroing out all the previous tokens.
> Hence, we argue the average pooling method is empirically and theoretically justified.
>
> **(6) L141-146 is not entirely accurate. Hard negative mining methods like [4] and [5] do not require this two-stage training procedure [which] is more of a recent trend in training powerful general-purpose embedding models.**
>
> We indeed ignored the existing approaches for mining the hard negatives. We have rewritten the mentioned paragraph as shown below (changes in bold) – if something is still imprecise, we would appreciate feedback!
>
> > […] which can be incorporated into the contrastive loss to promote learning more precise representations. **Moreover, there are works that focus on approaches for mining hard negatives which result in better training data in the context of specific downstream tasks [4,5].**
>
> > However, **recent works aiming at training powerful, general-purpose embedding models** that do rely on datasets with hard negatives often arrive at embeddings by having two distinct training phases […]
>
> ### References
>
> [1] Li et al.: Towards General Text Embeddings with Multi-stage Contrastive Learning. arXiv 2023\
> [2] Wang et al.: Improving Text Embeddings with Large Language Models. arXiv 2024\
> [3] Wang et al.: Text Embeddings by Weakly-Supervised Contrastive Pre-training. arXiv 2022\
> [4] Xiong et al.: Approximate Nearest Neighbor Negative Contrastive Learning for Dense Text Retrieval. ICLR 2021\
> [5] Zhan et al.: Optimizing Dense Retrieval Model Training with Hard Negatives. SIGIR 2021\
> [6] Kwiatkowski et al.: Natural Questions: A Benchmark for Question Answering Research. TACL 2019\
> [7] Hasibi et al.: DBpedia-Entity v2: A Test Collection for Entity Search. SIGIR 2017\
> [8] Muenninghoff et al.: MTEB: Massive Text Embedding Benchmark. EACL 2023

---

> > ### Author Response · Authors · 2024-08-11
> >
> > Again, we thank the reviewer for their constructive feedback. As we have passed the middle of the discussion period, we would like to confirm if our answers are comprehensive and satisfactory. Should any new questions arise, we would be happy to answer them.

---

### Official Review · Reviewer_GCkf · 2024-07-18

**Soundness:** 2
**Presentation:** 3
**Contribution:** 4
**Rating:** 6
**Confidence:** 3

**Summary:**

This paper focuses on the efficient contrastive training of text embedding models using pre-trained decoder-only language models. The main contribution is an algorithm that determines the best configuration of model sizes, data amounts, and fine-tuning methods for different computational budgets. Through extensive experimentation, the authors developed a guideline that helps practitioners choose the most suitable design options for their text embedding models. The study finds that full fine-tuning is best for lower computational budgets, while low-rank adaptation fine-tuning is optimal for higher budgets.

**Strengths:**

The research question posed, “What is the best embedding model one can train from a backbone decoder-only LLM with a fixed budget?” is highly valuable and important. It addresses a critical need within the NLP and IR domains.

The paper boasts a robust experimental setup, and the conclusions drawn from these experiments are insightful. The analysis based on empirical results provides a useful reference for advancements in both NLP and IR fields. This thorough approach enhances the reliability and applicability of the research findings.

**Weaknesses:**

The paper does not introduce a new methodology. The techniques employed, such as scaling laws, extracting representations from transformers, and contrastive fine-tuning, are all well-established in the literature.

There are several existing studies that the paper fails to acknowledge, which could provide a richer context and enhance the literature review. Notable omissions include works like “LLM-Oriented Retrieval Tuner” and “Scaling Laws for Dense Retrieval.” Acknowledging and discussing these related studies could strengthen the paper by positioning it within the existing research landscape more accurately.

**Questions:**

Please refer to the previous sections.

**Limitations:**

This paper discusses the limitation in Section 5.

---

> ### Author Rebuttal · Authors · 2024-08-06
>
> We thank the Reviewer for their effort to assess our work! Below we respond to the two points of critique.
>
> **(1) The paper does not introduce a new methodology. The techniques employed, such as scaling laws, extracting representations from transformers, and contrastive fine-tuning, are all well-established in the literature.**
>
> We agree with the Reviewer that our study does not introduce novel techniques – and this was not our intention in this project. We rather aimed to investigate and compare existing modern approaches in the practical, limited compute-budget setting. Before our work, no study had been devoted to establishing compute-efficient approaches to contrastive fine-tuning of decoder-only language models. Given the recent activity in this area, we found it to be an important research gap that needs to be addressed. We believe that our findings constitute a *concrete scientific contribution* to helping researchers and practitioners working with embedding models.
>
> **(2) There are several existing studies that the paper fails to acknowledge, which could provide a richer context and enhance the literature review. Notable omissions include works like “LLM-Oriented Retrieval Tuner” and “Scaling Laws for Dense Retrieval.”**
>
> Thank you for pointing out these two recent works! We have extended the related work section to include both of them. Specifically, starting from line 95 (new text in bold):
>
> > These approaches are all applicable to embedding models. Muennighoff [1] explored a simple parameter-efficient approach to repurpose GPT models into embedding models where only the bias tensors of the transformer model are updated [2]. **Sun et al. [3] developed a parameter-efficient method designed specifically for fine-tuning embedding models.** However, a systematic study of what parameter-efficient methods are optimal under what scenarios has not been performed, which is the aim of our work.
>
> Also, starting from line 84:
>
> > **The single most related investigation is a concurrent work [4], where the scaling laws for encoder models for retrieval are investigated. There are significant differences in the settings we consider. Our work focus is investigating the process of fine-tuning decoder-only models for good quality embeddings.** Moreover, our main goal is to find which strategy for achieving good embeddings is optimal in a budget-restricted setting, while taking into account the popularity of applying parameter efficient methods for fine-tuning, like LoRA or partial model freezing.
>
> ### References
>
> [1] Muennighoff: SGPT: GPT sentence embeddings for semantic search. arXiv 2022\
> [2] Zaken et al.: BitFit: Simple Parameter-efficient Fine-tuning for Transformer-based Masked Language-models. ACL 2022\
> [3] Sun et al.: LLM-Oriented Retrieval Tuner. arXiv 2024\
> [4] Fang et al.: Scaling Laws For Dense Retrieval. SIGIR 2024
>
> ---
> *We hope that we adequately addressed the Reviewer’s concerns. If that is the case, we kindly ask for a reconsideration of the paper score.*

---

> > ### Author Response · Authors · 2024-08-11
> >
> > Dear reviewer, we thank you again for your review. We would like to confirm if our rebuttal answers are satisfactory. Please let us know if you have any new questions.

---

### Author Rebuttal · Authors · 2024-08-07

We thank the Reviewers for their effort and constructive feedback. We believe that it will lead to a substantially better version of our paper.

We are pleased that the Reviewers found
* the research question our work addresses relevant and practical [GCkf, 4Bw9, zJJS, xguk],
* our experimental setup extensive and robust [GCkf, 4Bw9, xguk],
* the conclusions drawn insightful [GCkf, xguk].

We are also grateful for all the constructive concerns that have been raised and make a substantial effort to address them all. Below is a summary of the main points of our rebuttal, whereas detailed responses are provided below for each individual review.

---

## Summary of the rebuttal

* We provide responses – largely supported by new preliminary experimental results (see below) – concerning all the issues raised by the Reviewers, most importantly:
  * transferability of our conclusions to other models [zJJS],
  * omitting the explicit analysis of the data-constrained setting in our paper [4Bw9, zJJS],
  * the robustness of our evaluation [xguk, 4Bw9].
* We resolved to update our paper for the camera-ready version with new experiments, namely:
  * the results from training the Gemma 2b model,
  * the evaluation results on a more realistic retrieval dataset.
* We improved the text of the paper in a few places:
  * we added new related work (kindly pointed out by Reviewer GCkf),
  * we made the discussion about the hard negatives more accurate (per suggestion of Reviewer 4Bw9),
  * we extended our limitations section with a paragraph admitting that our study remains inference-cost-agnostic (thanks to the comment by Reviewer xguk).

---

## New experimental results

During the limited rebuttal time, we have managed to run a number of experiments regarding the raised concerns: transferability of our conclusions to other models [zJJS] and our MTEB subset choice for evaluation [xguk, 4Bw9]. We give details in the answers to the reviewers indicated in the brackets. In a nutshell:
* We found that for a new model – Gemma 2B – for two tested methods (full fine-tuning and LoRA), the results follow the same pattern as for our Pythia experiments.
* We found that there is a very strong correlation between our selected subset and the whole MTEB evaluation.
* We also confirmed that regarding the retrieval task specifically, the results on the SciFact benchmark that we used are highly correlated with much larger retrieval benchmarks.

---

We believe that this makes our paper stronger, and again, we are grateful for the Reviewers’ suggestions!

---

### Author Response · Authors · 2024-08-12
**New Results**

Dear Reviewers,

We are happy to announce one more result concerning the very recent Gemma-2-2b model (released two weeks ago). Despite the limited time, we managed to conduct experiments with full fine-tuning and observed that they were consistent with those obtained with Gemma-2 (see the first subplot of Figure 4 in the attached document). This result further supports the robustness and broad applicability of our work. We will present a more exhaustive study on this model in the camera-ready version.

Best regards,
Authors

---

### Decision · Program_Chairs · 2024-09-25

**Decision:**

Accept (poster)

**Comment:**

This work investigates how to contrastively train text embedding models in a compute-optimal manner using a suite of pre-trained decoder-only language models. The motivation is solid, and the quality of the work is high. However, one limitation is the relatively small model sizes used in the study, with a maximum of 3B parameters. It would be more compelling to explore the scaling laws for larger models, especially considering that the leading embedding models today are typically around 7B parameters e.g., NV-Embed, bge-en-icl, SFR-Embedding.
https://huggingface.co/spaces/mteb/leaderboard